# IL13 Promoter (−1055) Polymorphism Associated with Leukocyte Mitochondria DNA Copy Number in Chronic Obstructive Pulmonary Disease

**DOI:** 10.3390/cells11233787

**Published:** 2022-11-26

**Authors:** Shih-Feng Liu, Hui-Chuan Chang, Yu-Ping Chang, Ho-Chang Kuo, Yuh-Chyn Tsai

**Affiliations:** 1Department of Respiratory Therapy, Kaohsiung Chang Gung Memorial Hospital, Kaohsiung 833, Taiwan; 2Department of Internal Medicine, Division of Pulmonary and Critical Care Medicine, Kaohsiung Chang Gung Memorial Hospital, Kaohsiung 833, Taiwan; 3Medical Department, College of Medicine, Chang Gung University, Taoyuan 333, Taiwan; 4Department of Pediatrics, Kaohsiung Chang Gung Memorial Hospital, Kaohsiung 833, Taiwan

**Keywords:** IL13 polymorphism, COPD, leukocyte mitochondria DNA copy number

## Abstract

IL13 polymorphism is associated with chronic obstructive pulmonary disease (COPD). Patients with COPD have smaller numbers of mitochondria deoxyribonucleic acid copies (mtDNA-CN) than people without COPD do. However, whether IL13 polymorphism affects the mutation and recombination of mitochondria remains unclear. Data for patients with COPD and non-COPD were collected from Kaohsiung Chang Gung Memorial Hospital to enable a comparison of their leukocyte mtDNA-CN and the association of this information with IL-13 promoter (−1055) polymorphism. This study included 99 patients with COPD and 117 individuals without COPD. The non-COPD individuals included 77 healthy individuals that never smoked and 40 healthy smokers. The patients with COPD exhibited significantly lower mtDNA-CN than non-COPD did (250.34 vs. 440.03; *p* < 0.001); mtDNA-CN was particularly pronounced in individuals with the IL13 CC and CT genotypes compared with individuals with the TT genotype. When only individuals without COPD were considered and when all participants were considered, the differences in the mtDNA-CNs in individuals with the CC and CT genotypes were more significant than those in individuals with the TT genotype (448.4 and 533.6 vs. 282.8; *p* < 0.05 in non-COPD group); (368.8 and 362.6 vs. 249.6, *p* < 0.05 in all participants). The increase mtDNA-CN in the CC and CT genotypes was also more than that in the TT genotype in COPD patients, but showed no significance (260.1 and 230.5 vs. 149.9; *p* = 0.343). The finding shows that COPD is a mitochondria regulatory disorder and IL-13 promoter (−1055) polymorphism is associated with leukocyte mtDNA-CN. Developing COPD control methods based on mitochondrial regulation will be possible.

## 1. Introduction

Chronic obstructive pulmonary disease (COPD) is a type of chronic respiratory tract inflammation that causes irreversible or partially reversible airway obstruction [1]. COPD is caused by long-term exposure to toxic particles, which primarily occurs because of cigarette use. However, only about 20% of smokers have been given a diagnosis of COPD [2]. Some genes, including IL13 polymorphism, have been verified as related to COPD [3,4,5,6]. However, these results remain controversial in studies on different ethnic groups or research populations [7,8,9]. Prior to this study, we verified a correlation between IL13- promote (−1055) T allele and the development and severity of COPD in Taiwanese population [10]. Our previous data showed the T-allelic frequencies of the IL13 −1055 gene polymorphisms in COPD group are significantly higher than those in control group (18.8% versus 1.4%; *p* < 0.001; odds ratio [OR] = 29.3; 95% confidence interval [CI]: 5.9–145.3); and the frequencies of CT/TT genotypes in COPD group are significantly higher than those in control group (27.1% versus 2.8%; *p* < 0.001; OR = 20.0; 95% CI: 3.9–100.8); and IL13 −1055 T allele is the independent factor associated with forced expiratory volume in 1 s (*p* = 0.007) [10].

Reactive oxygen stress (ROS), a key factor in the pathological mechanism of COPD, induces chronic inflammation and cellular senescence, impairs autophagy, reduces deoxyribonucleic acid (DNA) repair, increases autoimmunity and mucus secretion, and impairs anti-inflammatory responses to corticosteroids [11]. The external causes of ROS include smoking, air pollution, and indoor cooking fire. The internal causes include the mitochondrial respiration of inflammatory cells. Mitochondria are eukaryotic organelles that generate energy. However, because mitochondria lack integrin and protected histone proteins, they are more likely to mutate and recombine than nuclear DNA is. Prior to this study, we verified that MtDNA copy number (mtDNA-CN) of peripheral leukocytes in the COPD group is significantly decreased compared with non-smoker group (250.3 ± 21.5 vs. 464.2 ± 49.9, *p* < 0.001). MtDNA copy number in the COPD group was less than that in the healthy smoking group, but *p* value nearly achieved significance (250.3 ± 21.5 vs. 404.0 ± 76.7, *p* = 0.08) [12]. The previous data also showed mtDNA-CN has no significance with age, gender, body mass index, current smoking, and pack-years in COPD group, healthy smoker group and no smoker group, respectively. Higher numbers of leukocyte mtDNA-CN can enhance the antioxidant properties of mitochondria. Accordingly, COPD is a disease associated with leukocyte mitochondria regulatory disorder.

IL13 frequently accompanies mitochondria-related conditions, including in infectious disease [13], inflammatory diseases [14], and metabolic diseases [15] and even in cancers [16]. IL13 has also been reported to be associated with mitochondrial dysfunction in an animal-induced asthma model [17]. The relationship between blood leukocyte mitochondrial DNA copy number and inflammatory cytokines, including IL13, was also investigated in knee osteoarthritis [18]. Update, IL13 is reported to be related to some mechanisms of COPD [19,20]. However, there is no report that IL13 in COPD is associated with mitochondrial dysfunction or mtDNA-CN. As mentioned above, IL 13 polymorphism and mtDNA-CN are both associated with the risk of COPD. In this study, we hypothesize that the association between IL13 promoter (−1055) polymorphism with the peripheral leukocyte mtDNA-CN in COPD.

## 2. Materials and Methods

### 2.1. Design

Two groups of participants included patients with COPD and non-COPD individuals, from the outpatient clinics of Chang Gung Memorial Hospital in Kaohsiung. The associations between peripheral leukocyte mtDNA-CN in the two groups and IL-13 sequencing genotypes were investigated.

### 2.2. Study Participants

The study participants including COPD and non-COPD group were enrolled from chest outpatient clinics of our hospital. The inclusion criteria and exclusion criteria of enrolled participants were described as below:

Patients who had been given a diagnosis of COPD were aged ≥ 40 years, had smoked for at least 10 pack-years, and exhibited the pulmonary airflow limitations outlined in the Global Initiative for Chronic Obstructive Lung Disease guidelines (a percentage predicted value for post-bronchodilator forced expiratory volume in 1 s [FEV1]/forced vital capacity [FVC] < 70% and four severe FEV1–air flow limitations) [1]. Airflow limitations caused by other factors were excluded according to the patients’ medical history and chest X-ray results.

The control groups were healthy individuals that never smoked and that smoked for at least 10 pack-years. The medical history, physical examination results, and pulmonary function test results of these individuals were carefully evaluated; FEV1/FVC and FEV1 were predicted to be >70% and >80%, respectively.

Participants with diseases associated with mtDNA-CN were excluded, namely those with a history of diabetes, liver disease, renal cell carcinoma, myocarditis, breast cancer, or chronic kidney disease.

### 2.3. Genotyping of IL13 Sequencing

Genomic DNA was extracted to enable genotyping of IL13 sequencing. Genomic DNA was extracted from 3 mL blood anticoagulated with Gentra PureGene ethylenediamine tetraacetic acid (EDTA; Gentra Biosystems, Minneapolis, MN, USA) according to the manufacturer’s instructions. The human IL13 reference sequence was obtained from the National Center for Biotechnology Information. To isolate the single nucleotide polymorphism at promoter position −1055 in the human IL13, polymerase chain reaction (PCR) was performed on a 50 µL solution of 50 ng DNA and 10 µM DNA polymerase premix of each primer (Yeastern Biotech, Taipei, Taiwan). The PCR products were purified using a DNA Clean/Extraction kit (GeneMark; Molecular Biology Tools, Taipei, Taiwan) and were cycle-sequenced using BigDye terminators (Applied Biosystems).

### 2.4. MtDNA-CN of Peripheral Blood Leukocyte

The mtDNA copy numbers were determined through quantitative real-time PCR assay by using DNA extracted from peripheral leukocytes. The contents of the mtDNA were determined in duplicate by using a 7900HT Sequence Detection System (Applied Biosystems). Gene-specific PCR primers were designed using Primer Express. The V.2.0 software (Applied Biosystems, Foster City, CA, USA) was developed using specific sequences from GenBank. The amount of mtDNA was expressed relative to the amount of RNAse P(2(−ΔCT)) and the ΔCT value calculated using Sequence Detector software, version 2.3.

### 2.5. Ethics

This study was approved by the Chang Gung Memorial Hospital Research Ethics Committee (IRB # 101–0260B). Written informed consent was obtained from all participants.

### 2.6. Statistical analysis

Continuous variables are presented as means ± standard deviation, and discrete variables are expressed as numbers and percentages. Categorical variables were compared using a chi-square test. Continuous variables were compared using a *t* test. The multiple comparisons among the three groups were analyzed through a one-way analysis of variance followed by post hoc corrections with the Tukey method. A *p* value < 0.05 was considered significant. SPSS, version 17, was used to perform all statistical tests.

## 3. Results

### 3.1. Patients’ Characteristics

This study included 99 patients with COPD and 117 individuals without COPD. The non-COPD individuals included 77 healthy individuals that never smoked and 40 healthy smokers. Clinical characteristics of COPD group and non-COPD group were shown in Table 1. Compared to non-COPD group, the COPD group were older (69.3 vs. 55.5 years), predominantly male (96 vs. 87%), more pack-years (50.94 vs. 36.75), lower FEV1 (56.8 vs. 93.9%), lower FEV1/FVC (53.17 vs. 83.82%), higher Modified Medical Research Council (mMRC) dyspnea scale 2.14 vs. 1.38), less six minute walk distance (6MWD) (437.28 vs. 520.17 m) and lower mtDNA-CN (250.34 vs. 520.17). The age, gender, BMI, pack-years, current smoking status are not associated with mitochondrial DNA copy number in COPD and non-COPD patients. The patients with COPD exhibited significantly lower mtDNA-CN than the individuals without did (*p* < 0.001) (Figure 1).

### 3.2. Comparison of mtDNA-CN in Individual IL 13 Genotype (CC, CT, TT) in COPD and Non-COPD Groups

For all IL13 genotypes, including CC, CT, and TT, non-COPD patients had significantly higher mtDNA-CN than COPD patients (Table 2 and Figure 2).

### 3.3. Comparison of Mitochondrial DNA Copy Number among IL 13 Genotypes (CC, CT, TT) in COPD Group, Non-COPD Group or Total Participants

As indicated in Table 3, regardless of whether only the patients with COPD, only the individuals without COPD, or all participants were considered, the increase mtDNA-CN in the CC and CT genotypes was more than that in the TT genotype, but only show significance in non-COPD patients and all participants (448.4 and 533.6 vs. 282.8; *p* < 0.05 in non-COPD group); (368.8 and 362.6 vs. 249.6, *p* < 0.05 in all participants). The increase mtDNA-CN in the CC and CT genotypes was also more than that in the TT genotype in COPD patients, but showed no significance (260.1 and 230.5 vs. 149.9; *p* = 0.343). (Table 3 and Figure 3).

## 4. Discussion

The results verified that COPD is a disease related to mitochondrial regulatory disorder and that IL13-promoted (−1055) polymorphism is associated with peripheral leukocyte mtDNA-CN. The level of mtDNA-CN was particularly pronounced in individuals with the IL13 CC and CT genotypes compared with individuals with the TT genotype. However, further investigations are required to elucidate the mechanism underlying the association. Although the increase in mtDNA-CN in the CC and CT genotypes was not significantly higher than that in the TT genotype in COPD patients (260.1 and 230.5 vs. 149.9; *p* = 0.343), there was a trend in IL13 CC and CT genotypes for mtDNA-CN was higher than TT genotype. Maybe we can collect more cases to reach statistical significance.

### 4.1. COPD Is Associated with Mitochondrial Dysfunction

Mitochondria are widely regarded as a primary internal source of ROS. In normal oxidative metabolism, mitochondria play a key role in cell signaling and homeostasis. Normal oxidative metabolism requires optimal levels of ROS. However, excessive ROS can be detrimental to cell integrity and damage lipids, proteins, and DNA. An increasing number of studies have reported that COPD is associated with mitochondrial dysfunction, which may involve impaired autophagy, reduced autophagic flux, and abnormal mitochondria accumulation [11,21,22,23]. COPD causes airway and quadricep mitochondrial dysfunction, and its development and severity are related to airway obstruction and exercise capacity. Mitochondrial dysfunction is frequently a therapeutic target in COPD treatment [24].

### 4.2. Association between IL13 and COPD

IL13 is a type of T-help 2 type cytokine and plays a key role in the pathophysiology of asthma and COPD. A mouse model of chronic type 2 pulmonary inflammation exhibited airspace enlargement dependent on MMP-12 and eosinophil-derived 13 [19]. Some IL13 polymorphisms increase the expression of IL13 messenger ribonucleic acid and protein in the blood, bronchial washes, and sputum of patients with asthma or COPD [20]. Anti-IL-13 therapies may be a novel therapy for chronic airway inflammatory disease.

### 4.3. Association between IL13 and Mitochondrial Regulation

Studies have reported that IL13 is associated with mitochondrial dysfunction in patients with sialadenitis [14] and protects the mitochondria of cardiomyocytes in patients with septicemia [13]. Increased IL13 expression has been reported in the dysfunctional mitochondria of mice with house dust mite-induced asthma [17]. IL13 has been reported to increase the glucose tolerance and mitochondrial activity of the muscles after exercise [15]. Mitochondrial oxidative stress in cancer cells has been indicated to induce an increase in IL13 [16]. IL-13 mediates ROS generation through activation of NADPH oxidase, consequently contributing to the degeneration of hippocampal neurons in vivo [25]. IL13-1055 polymorphism leads to different levels of IL13 secretion, which lead to different mitochondrial activity in smokers. Additionally, the functional state of mitochondria depends on the balance between their biogenesis and degradation (mitochondrial autophagy). Dysfunctional mitochondria are a source of oxidative stress and inflammasome activation. In COPD, upon activation of the Pink1 mitophagy pathway, the translocation of the ubiquitin-related degradation molecule Parkin is impaired, resulting in mitochondrial dysfunction [26]. In this study, we speculate that the mtDNA-CN associated with IL13 production in patients with TT genotype is lower than that in patients with CC and CT genotypes, or the mitochondrial repeat mutation and recombination ability of TT genotype are poorer. Consequently, the TT genotype accumulates the most ROS and, therefore, leads to the highest risk of COPD infection.

This study has several limitations. We used specimens that remained from our previous studies to obtain IL13-promoter (−1055) genotype analysis data. However, we were unable to determine the IL13 concentrations in the serum. Therefore, we were unable to analyze serum IL13 levels, IL13 genotypes, differences between people with and without COPD, and the association between these factors and mtDNA-CN. Moreover, IL13 polymorphism exhibited a stronger association with COPD in prior studies than it did in this study. Additionally, due to lack of research funding, the number of samples used in this study was limited; only available data were analyzed. Furthermore, since this is a candidate gene study and only the IL13 polymorphism was sequenced, a more comprehensive genome-wide association study is required.

## 5. Conclusions

COPD is a mitochondrial regulation disorder. The patients with COPD exhibited significantly lower mtDNA-CN than the individuals without COPD. Developing COPD new therapy based on mitochondrial regulation will be possible. Additionally, IL13-promoted (−1055) polymorphism may be associated with mtDNA-CN. The increase mtDNA-CN in the IL13 CC and CT genotypes was more than that in the TT genotype in our study. Due to the small participants, the mitochondrial regulatory mechanism of COPD potentially related to IL13 still needs more research to explore.

## Figures and Tables

**Figure 1 cells-11-03787-f001:**
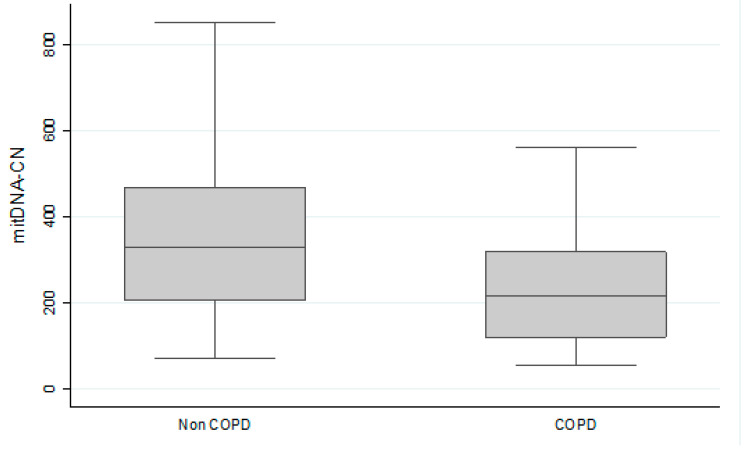
The patients with COPD exhibited significantly lower mitochondria DNA copy number than the individuals without did. (*p* < 0.05).

**Figure 2 cells-11-03787-f002:**
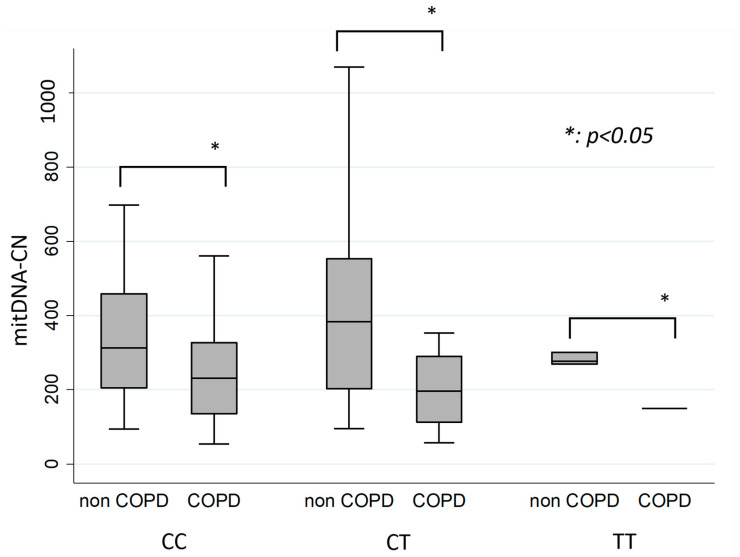
Comparison of mitDNA-CN in individual IL 13 genotype (CC, CT, TT) in COPD and non-COPD groups.

**Figure 3 cells-11-03787-f003:**
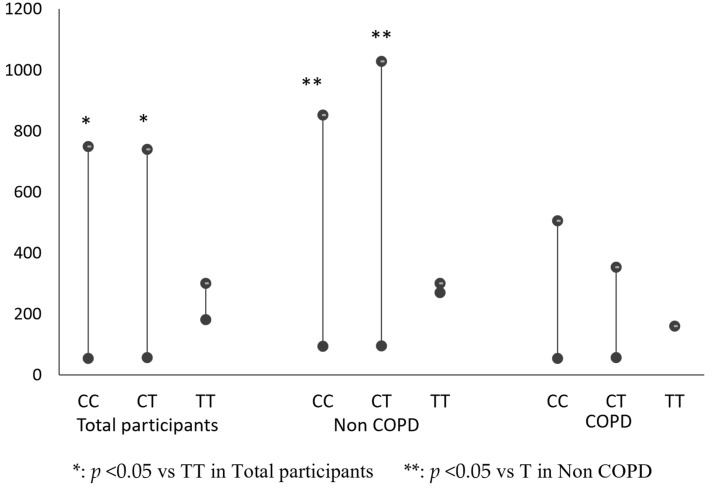
Comparison of mitochondrial DNA copy number among IL 13 genotype (CC, CT, TT) in COPD group, non-COPD group or total participants.

**Table 1 cells-11-03787-t001:** Clinical characteristics of COPD group and non-COPD group.

	Non COPD (*n* = 117)	COPD (*n* = 99)	*p* Value
	Mean (SD)	Mean (SD)	
Age	55.50 (9.67)	69.34 (11.043)	<0.001
Gender (male, %)	87	96	0.023
Pack-years	36.75 (30.59)	50.94 (31.19)	0.014
BMI	24.43 (3.33)	23.67 (4.01)	0.140
FVC	89.95 (11.63)	80.80 (12.97)	<0.001
FEV1	93.91 (1.35)	56.82 (19.93)	<0.001
FEV1/FVC	83.82 (7.78)	53.17 (11.29)	<0.001
mMRC	1.38	2.14	0.003
6MWD	520.17 (134.14)	437.28 (109.91)	0.018
mtDNA-CN	444.03 (433.08)	250.34 (199.75)	<0.001

mtDNA-CN: mitochondria DNA copy number; BMI: Body mass index; mMRC: Modified Medical Research Council; 6MWD: Six minute walk test; FVC: Forced vital capacity; FEV1; Forced expiratory volume in 1 s; mtDNA-CN: mitochondria DNA copy number; COPD: chronic obstructive pulmonary disease.

**Table 2 cells-11-03787-t002:** Comparison of mitochondrial DNA copy number in individual IL 13 genotype (CC, CT, TT) in COPD and non-COPD groups.

	Non COPD (*n* = 117)		COPD (*n* = 99)		
IL13 Genotypes	N	mitDNA Copy Number Mean ± SD	N.	mitDNA Copy Number Mean ± SD	*p* Value
Total	117	444.03 ± 433.08	99	250.34 ± 199.75	0.000
CC	92	451.40 ± 455.35	71	260.10 ± 208.32	0.0038
CT	21	533.57 ± 494.72	27	230.48 ± 174.37	0.0111
TT	4	282.79 ± 16.29	1	149.85 ± 0	0.0194

COPD: chronic obstructive pulmonary disease; mtDNA-CN: mitochondria DNA copy number.

**Table 3 cells-11-03787-t003:** Comparison of mitochondrial DNA copy number among IL 13 genotypes (CC, CT, TT) in COPD group, non-COPD group or total participants.

	CC N(%)	mtDNA-CN Mean ± SD	CT N(%)	mtDNA-CN Mean ± SD	TT N(%)	mtDNA-CN Mean ± SD	*p* Value
COPD	71(71.6%)	260.10 ± 208.32	27(27.2%)	230.48 ± 174.37	1 (1.2%)	149.85 ± 0.00	0.343
Non-COPD	92(78.7%)	448.37 ± 457.69	21(18.1%)	533.57 ± 494.72	4(3.2%)	282.79 ± 16.29	0.005
Total participants	163(75.6%)	368.80 ± 380.42	48(22.2%)	362.59 ± 378.20	5(2.3%)	249.56 ± 67.79	0.030

COPD: chronic obstructive pulmonary disease; mtDNA-CN: mitochondria DNA copy number.

## Data Availability

The data generated and analyzed in this study are included in the article.

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
