# Peer review of "IL13 Promoter (−1055) Polymorphism Associated with Leukocyte Mitochondria DNA Copy Number in Chronic Obstructive Pulmonary Disease"

_cells, 2022, doi:10.3390/cells11233787_

Round 1
Reviewer 1 Report
The title of the article fully reflects the content of the article.
The abstract briefly outlines the information necessary for the reader: the relevance of the study, the purpose, the object of the study. From the presented material, the authors' intention and the design of the study are clear. The section presents the main results of the study. The novelty of the results is indicated. The novelty of the research is absolute. The conclusion is concrete and presented clearly, logically follows from the results. Although the authors did not indicate this, the potential for practical application of the results is clearly visible: the possibility of developing COPD control methods based on mitochondrial regulation.
The abstract would be more complete if the authors indicated the groups of participants and the studied population of people. In addition, the idea would be more understandable if, in the abstract, the authors would point to leukocytes as the cells that were studied in this work.
The presented keywords fully reflect the research topic presented by the authors and are necessary.
In the "Introduction" section, the authors briefly, about nearly enough for the reader, characterized chronic obstructive pulmonary disease (COPD). The authors presented data on the T-allele frequency of IL13 -1055 gene polymorphism in the COPD group and healthy volunteers. The authors identified active oxygen stress as a key factor in the pathological mechanism of COPD, which disrupts, among other things, mitochondrial respiration due to mutations. According to the authors, COPD is a disease associated with a violation of the regulation of the mitochondria of leukocytes. The aim of the study is clear and is to confirm the hypothesis of the association of IL13 (-1055) promoter polymorphism with mtDNACN of peripheral leukocytes in COPD.
The section "Materials and methods" presents the general design of the study. The design of the study is clear. The authors identified the groups of participants and the parameters of the study. Groups of participants (for example, healthy people and patients with COPD) were characterized. The criteria for exclusion from the study are presented. Patients with COPD were characterized according to the recommendations of the Global Initiative on Chronic Obstructive Pulmonary Disease. This study was approved by the Research Ethics Committee of Chang Gung Memorial Hospital. Written informed consent was received from all participants. The section would be more complete if the authors indicated the studied population of study participants.
The section presents the main methods that allowed the authors to achieve the goal of the study: Genotyping of IL13 sequencing, MtDNACN of peripheral blood leukocytes. The selected statistical tests correspond to the purpose of this study.
In the section "Obtained results", the authors consistently demonstrate the results of their research. All the presented results of the study relate to COPD. The clinical characteristics of patients with COPD are presented and the number of copies of mitochondrial DNA in patients with COPD and in persons without COPD is shown. The authors compared mtDNACN in the individual IL 13 genotype (CC, CT, TT) in the groups with and without COPD, as well as the number of copies of mitochondrial DNA among the IL 13 genotypes (CC, CT, TT) in the group with COPD, the group without COPD, or in the total number of participants.
In the Discussion section, the authors presented literature data confirming the association of COPD with mitochondrial dysfunction, the relationship between IL13 and COPD, as well as the interaction of IL13 and mitochondrial regulation. These data made it possible to discuss the results obtained from various points of view. The authors rightly pointed out a number of limitations of this study. Due to these limitations, the authors were unable to analyze serum IL13 levels, IL 13 genotypes, differences between people with and without COPD, as well as the relationship between these factors and mtDNA-C. These limitations indicate the need for future research within the scope of the topic of the article.
The authors' conclusion fully corresponds to the results obtained.
All tables are clear and legible, and are necessary to understand the results of the study. The drawings complement the article.
The article does not cause any concerns. The manuscript did not cause any ethical problems. Statistical analysis corresponds to the study. All references to publications presented by the authors in the article are necessary and correct, made in the right style. Of the 28 links that are presented in the article, 14 links are from the last 5 years (2017-2022). Pay attention to the links under the number 10-11. There is a technical error here. Please correct this (these) links. I have no concerns about the similarity of this article with other articles published by the same authors.
Competing interests of the authors do not create bias in the presentation of results and conclusions.
Reviewer 2 Report
There is only one clear result in this study: COPD subjects have significantly fewer mtDNA-CNs than non-COPD subjects. Another significant finding is the presence of fewer mtDNA-CNs in the TT genotype than in the CC and CT genotypes. However, this result is to be considered with great caution given the very small number of subjects with TT genotype studied (5). Even weaker is the association between mtDNA-CNs, the TT genotype, and the presence of COPD, as there is only one subject with COPD and TT genotype. In conclusion, I agree with the authors when they write that it would be necessary to study a higher number of subjects in order to have really significant data beyond the result of a single statistical test.
One minor observation: please avoid to report the mean and standard deviation for the mMRC score. I recognize that this is a quite common reporting method for this variable, but, in reality, mMRC is an ordinal variable and should be reported using categories, and evaluated with proper statistical methods, if necessary.
Reviewer 3 Report
This article discusses COPD disease that is related to IL 13 polymorphism.
The subject seems interesting but the description is quiet vague. For instance, I missed the actual number of patients aided in this study. The authors mention "3 groups" without mentioning the tital number of population that is very important in a statistical study like this.
In addition, I miss a real conclusion for the study, although the authors exerted considerable effort to display the results to their best possible. Some refinements and accuracy would turn this article to a much better version of itself.
Round 2
Reviewer 2 Report
I still remain unsatisfied with your answer to my minor observation. The MMRC dyspnea score is still reported as mean value. However, it should be reported using the frequency of each category or, to save space, reducing the five categories of the MMRC score to two only. For example, according to the suggestion of the GOLD initiative, frequency of subjects with MMRC >=2